# The Tipping Effect of Delayed Interventions on the Evolution of COVID-19 Incidence

**DOI:** 10.3390/ijerph18094484

**Published:** 2021-04-23

**Authors:** Kristoffer Rypdal

**Affiliations:** Department of Mathematics and Statistics, UiT—The Arctic University of Norway, 9019 Tromsø, Norway; kristoffer.rypdal@uit.no

**Keywords:** COVID-19, epidemic curve, epidemic waves, reproduction number, social-response model, delayed response, tipping point, delay differential equations, vaccination, post-pandemic normal

## Abstract

We combine infectious disease transmission and the non-pharmaceutical (NPI) intervention response to disease incidence into one closed model consisting of two coupled delay differential equations for the incidence rate and the time-dependent reproduction number. The model contains three parameters, the initial reproduction number, the intervention strength, and the response delay. The response is modeled by assuming that the rate of change of the reproduction number is proportional to the negative deviation of the incidence rate from an intervention threshold. This delay dynamical system exhibits damped oscillations in one part of the parameter space, and growing oscillations in another, and these are separated by a surface where the solution is a strictly periodic nonlinear oscillation. For the COVID-19 pandemic, the tipping transition from damped to growing oscillations occurs for response delays of about one week, and suggests that, without vaccination, effective control and mitigation of successive epidemic waves cannot be achieved unless NPIs are implemented in a precautionary manner, rather as a response to the present incidence rate. Vaccination increases the quiet intervals between waves, but with delayed response, future flare-ups can only be prevented by establishing a post-pandemic normal with lower basic reproduction number.

## 1. Introduction

A year after the COVID-19 pandemic began its rapid geographic expansion across the globe, it is evident that the incidence rate in each country evolves in waves. In Europe, the typical pattern so far has been two waves, the first in the spring of 2020, and a second longer and stronger wave that started in the fall and is still ongoing at the time of writing, spring 2021 [1]. The increased strength of the second wave is particularly prominent in the case notification rate, but part of this increase is due to increasing testing rate. Nevertheless, the tendency is clear also in the reported COVID-19 death rates, although many countries have seen lower death rates in the start of the second wave, because this wave began with infection spreading in the younger age groups.

Based on the experience from the first wave, countries should have been better prepared for the second wave than for the first, and mitigation should have been be more feasible. There is no apparent microbiological mechanisms that could have driven the strong second wave, even though new and more contagious mutants have started to make an impact as we enter the calendar year of 2021. Hence, the explanation seems to be associated to how our societies respond to the threats of the pandemic.

Some evidence suggest that the wavy pattern during the first year of the pandemic has been driven by an interaction between the pathogen’s natural tendency to reproduce and the non-pharmaceutical interventions (NPIs) implemented by governments [2]. Even though the emergence of new and more contagious mutants of the SARS-CoV-2 virus with higher basic reproduction numbers R and the roll-out of vaccines will play an increasingly important role in reducing the effective reproduction numbers $R_{\mathrm{eff}}$, NPIs are expected to play an important role in regulating and controlling the incidence rate also in the upcoming year.

In this paper the term incidence rate X(t) will refer to the daily number of actual infections taking place in a country at the time *t*. It does not refer to the recorded incidence (case notification rate), and the time *t* is the time of infection, not the time the infection is detected. The instantaneous basic reproduction number R(t) refers to the average number of new infections transmitted by an infected individual at the time *t* in a population where all individuals are susceptible to the infection. The effective reproduction number $R_{\mathrm{eff}}\left(t\right)$ is the reproduction rate in a population where only a fraction S(t) is susceptible at time *t*.

There is clear and strong correlation between case notification rate and NPIs in most countries, and the time lag between NPIs and changes in recorded incidence corresponds roughly to the sum of incubation period and time for testing, analysis and registration. Thus, it is reasonable to assume that the effect of NPI-induced changes of R(t) on the actual transmission of the infection is more or less instantaneous. The same is not the case with the effect of disease incidence on the NPIs. Here we would expect considerable delay between cause and effect, a delay we shall refer to as the social response time.

NPIs represent a great burden on society, and so far in the COVID-19 pandemic there are very few examples where interventions have been effectuated in a precautionary manner. Political pressure has forced policy makers to respond to the *recorded* disease burden, which is delayed by 1–2 weeks relative to the actual state, even though most governments have access to model projections that can inform them about the true present state of the epidemic and the likely development in the near future. The objective of this paper is to investigate whether or not this delay may have an important influence on the trajectory of the epidemic state.

It is intuitively evident that NPIs effectuated as responses to the true epidemic state will lead to oscillations in the disease incidence. This is because NPIs act as a restoring force counteracting the virus’ natural tendency to reproduce, while the disease activity level below or above a socially acceptable threshold will enhance or reduce the NPIs. In a recent paper [2], we constructed a simple model that reduces to a damped harmonic oscillator in the small-amplitude (linearized) limit. In that paper we demonstrated that a weakening of the intervention response over time could counteract the damping and lead to stronger and longer secondary waves, but it was assumed that the intervention response is instantaneous. In the present paper, I explore a similar model, where the intervention fatigue is replaced by a delayed response.

In Section 2, I formulate and explain the mathematical model, which takes the form of a system of first-order delay differential equations [3], and we discuss briefly the nature of the equilibria and a possible limit cycle of the system and their relation to three model parameters expressing the reproductive ability of the pathogen, the intervention strength, and the response delay. I also introduce the effect of immunization through vaccination in the model. Then I explore the solutions of the system numerically in Section 3, and demonstrate the existence of a tipping transition that transforms the solution from damped into growing oscillations, and I map the surface in the parameter space where this transition takes place. I explore the effect of vaccination programs on this behavior and also the necessity of establishing a post-pandemic normal with a lower permanent basic reproduction number to prevent future flare-ups of the pandemic. In Section 4 I discuss the possible policy implications of these results.

## 2. Methods

### 2.1. Evolution of Epidemic State under Given Social Evolution

Let J(t) be the cumulative fraction of infected individuals in a population, and I(t) the instantaneous fraction of infectious individuals. The time evolution of these quantities can be modeled by the simple system of ordinary differential equations,
(1)$$\begin{array}{ccc} d_{t}J\left(t\right) & = & \alpha R_{\mathrm{eff}}\left(t\right)I\left(t\right), \end{array}$$
(2)$$\begin{array}{ccc} dI_{t}\left(t\right) & = & \alpha [R_{\mathrm{eff}}\left(t\right)-1]I\left(t\right)\;. \end{array}$$

Here the notation $d_{t}$ stands for the derivative with respect to time, $R_{\mathrm{eff}}\left(t\right)$ is the effective reproduction number at time *t* and $\alpha^{-1}$ is the mean duration of the infectious period. The system is a reformulation of the standard Susceptible-Infectious-Recovered (SIR) model of Kermack and McKendrick [4]. Here, the effective reproduction number can be written in the form,
(3)$$R_{\mathrm{eff}}\left(t\right)=R\left(t\right)S\left(t\right),$$
where R(t)=β(t)/α is the basic reproduction number which measures the average number of new infections transmitted by one infected individual if the entire population is susceptible to the infection, S(t)=1−J(t) is the fraction of susceptible individuals in the population, and β(t) is the contact rate. An important point is that α is time-independent and determined by the pathogen, β(t) is completely determined by the contagiousness of the pathogen and the evolution of the social state, while S(t) depends on the degree of immunity in the population. In the part of this paper which does not include the effect of vaccination, we shall assume that the degree of infection-induced immunity does not change significantly during the time span of the study, implying that we can consider S≈1, and hence that $R_{\mathrm{eff}}\left(t\right)\approx \left(\beta (t)/\alpha \right)=R\left(t\right)$ only varies in time due to variations in the social conditions that determine the contact rate β(t).

Note also that $\alpha [R_{\mathrm{eff}}\left(t\right)-1]$ is the relative growth rate for the infectious fraction, $\gamma_{I}\left(t\right)\equiv d_{t}\ln I(t)$, which is positive when $R_{\mathrm{eff}}\left(t\right)>1$ and negative when $R_{\mathrm{eff}}\left(t\right)<1$.

### 2.2. Evolution of Social State under Given Epidemic Evolution

Equations (Equation 1)–(Equation 3) describe the dynamics of the epidemic state J(t) and I(t) when the susceptible fraction S(t) is known and the evolution of the social state represented by R(t) is given. Assuming for the time being that S(t) is known, a closed model can be obtained if we can find an equation that connects R(t) to J(t) and I(t). This requires a description of how the social contact rate responds to the epidemic state. We shall represent this response by assuming that the relative rate of change $\gamma_{R}\equiv d_{t}\ln R\left(t\right)$ is a linear function of the delayed incidence rate $X\left(t-d_{t}\right)\equiv d_{t}J\left(t-t_{d}\right)$, where $t_{d}$ is the time delay. This function is positive when $X(t-t_{d})$ is below a threshold $X^{*}$ and negative when it is above that threshold. Society reacts to the incidence rate only when it receives the information about new infections, which is the main reason for the delay. When the incidence rate is low, society responds by relaxing restrictions, and the reproduction number increases. When the incidence rate exceeds the threshold $X^{*}$, restrictions are introduced that make $d_{t}\ln R\left(t\right)$ to change sign from positive to negative. Thus, we end up with the equation,
(4)$$d_{t}\ln R\left(t\right)=-k\left[X\left(t-t_{d}\right)-X^{*}\right],$$
where *k* is a coefficient which characterizes the strength of the social response to the epidemic evolution, and we shall refer to it as the intervention strength parameter.

One caveat of this formulation is that it necessarily fails if the model at some point in time predicts that R exceeds its “natural” value $R_{N}$ for the virus in a fully susceptible population with no NPIs implemented. If R exceeds $R_{N}$ the right hand side of Equation (Equation 4) should quickly drop to zero, because it is not reasonable to expect that society will implement measures that stimulate to higher reproduction numbers than we would have without any measures. One way to account for this in Equation (Equation 4) is to let the coefficient *k* depend on R(t) is such a way that it quickly drops to zero when R exceeds $R_{N}$. In practice we do this by putting
(5)$$k\left(R\right)=\left(1/2\right)\left(1-\tanh\left(R-R_{N}\right)\right)k_{0}.$$

The function $k\left(R\right)/k_{0}$ is plotted in Figure 1a for $R_{N}=4.0$.

### 2.3. A Closed Model for the Socio-Epidemic State

In the following, it is convenient to introduce a dimensionless time variable t→αt, which allows us to formulate the differential equations as functions of time measured in units of the infectious time $\alpha^{-1}$, rather than days. We also express the incidence rate and the infectious fraction in units of the intervention threshold, i.e., we introduce the dimensionless variables $X\left(t\right)\to X\left(t\right)/X^{*}$ and $I\left(t\right)\to I\left(t\right)/X^{*}$. Equation (Equation 1) can then be written as $X\left(t\right)=R_{\mathrm{eff}}\left(t\right)I\left(t\right)$ and inserted into Equation (2), which leaves us with the following nonlinear system of delay differential equations;
(6)$$\begin{array}{ccc} d_{t}\ln R\left(t\right) & = & -\kappa \left(R\left(t\right)\right)\left[R_{\mathrm{eff}}\left(t-\delta \right)I\left(t-\delta \right)-1\right],\\ d_{t}I\left(t\right) & = & [R_{\mathrm{eff}}\left(t\right)-1]I\left(t\right), \end{array}$$
where $\kappa \left(R\right)=k\left(R\right)X^{*}/\alpha$ and $\delta =t_{d}\alpha$. Since this is a system of delay differential equations we have to specify the state variables in the time interval t∈(−δ,0) rather than only at the time t=0 as in a conventional initial value problem. For this particular problem, we can do this in a way that reflects the actual epidemiological situation. In the early stage of the epidemic, the reproduction number is $R_{0}$ which is determined by the infectivity of the pathogen and the social structure in the actual country in absence of any non-pharmaceutical interventions. Let us define the time origin t=0 as the time when interventions start. At this time we assume S(0)=1. In in the model system (Equation 6) the threshold for intervention is X=RI=1, but because of the delay, the intervention that starts to change R at t=0 is a response to the reported incidence rate which took place at t=−δ, which means that X(−δ)=1. Since $R\left(t\right)=R_{0}$ for t∈(−δ,0) we have that $X\left(-\delta \right)=R_{0}I\left(-\delta \right)=1$, i.e., $I\left(-\delta \right)=1/R_{0}$. Equation (Equation 6) is valid not only for t>0, but also in the time interval (−δ,0) when $R\left(t\right)=R_{0}$, so the solution for I(t) satisfying the condition $I\left(-\delta \right)=1/R_{0}$ in this interval yields the following “initial conditions” for the interval t∈(−δ,0);
(7)$$R\left(t\right)=R_{0},\;\;I\left(t\right)=\left(1/R_{0}\right)\exp\left[\left(R_{0}-1\right)\left(t+\delta \right)\right].$$

Note also, that this choice does not only make epidemiological sense, but also ensures continuity in the derivatives of R(t) and I(t) across the intervention point t=0.

The system has two equilibrium states: a fixed point in $R_{\mathrm{eff}}=1$ and I=1, where the number of infected stays constant at the threshold value, and another fixed point in $R_{\mathrm{eff}}=0$ and I=0, which is a state with no transmission and nobody infected. The latter is obviously a repellor: if $R_{\mathrm{eff}}\left(t\right)$ and I(t) both are becoming very small, then the second equation in (Equation 6) implies that I(t) decays exponentially towards zero as I(t)≈exp(−t), while the first equation implies that $R_{\mathrm{eff}}\left(t\right)\approx \exp\left(\kappa t\right)$ grows exponentially.

The numerical exploration, for which results are presented in Section 3, demonstrates that the equilibrium $\left(R_{\mathrm{eff}},I\right)=\left(1,1\right)$ is a stable spiral node for some regions of the parameter space $\left(R_{0},\kappa_{0},\delta \right)$, it is an unstable spiral node in another region, and these regions are separated by a surface in parameter space where the solution is a limit cycle.

### 2.4. The Effect of Vaccination-Induced Immunity

At the time of writing, vaccination programs are rolled out at varying pace in most countries that have been severely hit by the pandemic, and in these countries vaccination immunity seems to be more important than immunity caused by infection. Vaccination reduces the susceptible population to $S_{V}\left(t\right)$, and the result is a lower effective reproduction number
(8)$$R_{\mathrm{eff}}\left(t\right)=S_{V}\left(t\right)R\left(t\right),$$
where I shall employ the following simple model for the susceptible fraction;
(9)$$S_{V}\left(t\right)=0.2+\frac{0.8}{1+{\left(t/\tau_{V}\right)}^{2}}.$$

The parameter $\tau_{V}$ is the time for completing half of the vaccination program, and Equation (Equation 8) implies that 20% of the population remains susceptible as $t/\tau_{V}\to \infty$. In Section 3 we shall assume that $\tau_{V}=10$ (or about 10 weeks). The function $S_{V}\left(t\right)$ is shown in Figure 1b.

The equation system (Equation 6) includes vaccination through $R_{\mathrm{eff}}\left(t\right)$ on the right hand side of both equations. This is because the rate of new infections is proportional to the number of susceptible individuals. On the left hand side of the first equation, however, we retain the relative growth rate of the *basic* reproduction number $d_{t}\ln R$, because it is R(t) that responds to the NPIs.

### 2.5. Numerical Exploration of the Parameter Space

Delay differential equations are integrated numerically by the same methods as ordinary differential equations, and these fast routines can be used to explore the nature of the solutions in the regions of the parameter space which is of interest to the COVID-19 pandemic. In particular, the region close to the transition surface is carefully mapped and we can easily detect the transition points with high accuracy in those three parameters. In practice, we run the routine for an array of values of the parameters $R_{0}$ and $\kappa_{0}$ and detect the value of δ for which the solution shifts from a decaying oscillation to a growing one.

## 3. Results

The delay-differential equation system (Equation 6), with “initial condition” (Equation 7), is an interesting object that warrants further mathematical study. The purpose of this paper, however, is not a mathematical exploration, but to extract those properties of the system that are of particular relevance to the delayed social response to changing reported incidence of the infection with the SARS-CoV-2 virus.

The introduction of dimensionless, independent and dependent variables has revealed that the epidemic evolution depends on the response time delay δ measured in units of the effective infectious time $\alpha^{-1}$. For COVID-19, a reasonable estimate of $\alpha^{-1}$ is about one week [2], which means that time in the plots in this paper is measured in weeks. It is reasonable to expect delays of the order of one week, so delay times in a range around δ∼1 are explored.

The interpretation of the dimensionless intervention strength κ can be seen from the first equation in (Equation 6), if we consider a state with very low incidence I(t)≪1, such that the first term on the right hand side can be neglected. In that case, we can write $\kappa \approx d_{t}\ln R$, which is the relative rate of change of R(t) measured in the time unit of a week. From experience with the first wave of the COVID-19 epidemic, we have seen that a characteristic time scale of change of R varies from a few weeks to a few months, which is included in the $\kappa_{0}$-range we explore below; $\kappa_{0}\in \left(0.2,1.0\right)$.

### 3.1. Solutions without Vaccination

Without vaccination, $S_{V}\left(t\right)=1$, and hence $R_{\mathrm{eff}}\left(t\right)=R\left(t\right)$. The three characteristic modes of epidemic development are shown in Figure 2. On a “transition surface” in the $\left(R_{0},\kappa_{0},\delta \right)$ parameter space, the solution to the system (Equation 6) is a nonlinear oscillation (limit cycle), as shown in Figure 2a,b, but this oscillation is parametrically unstable. This means that an infinitesimal perturbation of the parameters could lead either to a damped oscillation, as in Figure 2c,d, or to a growing oscillation, as in Figure 2e,f, depending on which side of the transition surface the perturbed parameter point is located.

The simulations shown in Figure 2 have been made with $R_{N}=4.0$, which is a reasonable basic reproduction number in the absence of NPIs for the more aggressive mutants circulating in the first months of 2021. However, since R in these simulations always stays well below $R_{N}$, there there is no significant effect of introducing the cut-off at $R=R_{N}$ through Equation (Equation 5). In Section 3.2 and Section 3.3, however, it will become clear that this cap $R_{N}$ on the basic reproduction number will play an important rôle when vaccination is introduced.

Figure 3 shows isolines (curves of constant $R_{0}$) in the $\left(\kappa_{0},\delta \right)$-plane for $R_{0}=1.1,1.7,2.3,3.0$. For any value of $R_{0}$ in this range and $\kappa_{0}$ in the range (0.2,1.0), the solution is a growing oscillation if the $\left(\kappa_{0},\delta \right)$-point is located above the isoline in the ($\kappa_{0},\delta$)-plane, and a damped oscillation below this line.

The take-home message from this figure is that there is a transition from a series of damped epidemic waves to a series of growing waves as the response delay δ exceeds a critical transition threshold ∼1, or in dimensional units, about one week. Thus, this simple model suggests that a policy responding blindly to the actual incidence rate delayed by more than approximately a week may lead to a succession of epidemic waves of increasing amplitude.

### 3.2. Solutions with Vaccination

At the time of writing, April 2021, a very low fraction of the world population has achieved immunity due to past infections, and even though vaccines are being rolled out at accelerating pace, there is growing concerns that herd immunity may never be attained [5]. There are several reasons for this concern. It is not known how long natural immunity from past infections will last, in particular considering the pace at which new mutants of the virus appears. It is not known how well, and how long, the vaccines protect against transmission. And it is not known whether children will be vaccinated at large scale and how large fraction of the population that will reject to take the vaccine. For these reasons, it is difficult to make precise and reliable estimates of the evolution of community immunity in the years to come. Nevertheless, it seems reasonable to assume that natural immunity due to past infections will play a minor rôle. Even if a majority of the population finally may have contracted the virus, waning immunity against new mutants will make such immunity insignificant compared to immunity obtained by new vaccines tailored to combat these new variants. In order to illustrate the possible effect of the vaccination programmes, I shall employ the model Equation (Equation 9) for the fraction of susceptible individuals $S_{V}\left(t\right)$, which is shown in Figure 1b. The results for parameters that are otherwise similar to those in Figure 2 are shown in Figure 4. In Figure 4a,b, I have chosen δ=1.1 to illustrate the convergence towards a stationary cycle, analogous to Figure 2a,b. Figure 4c,d, for δ=0.7, show an example of a decaying oscillation that will finally end up in the equilibrium point $\left(R_{\mathrm{eff}},X\right)=\left(1,1\right)$, and Figure 4e,f, for δ=1.1, display growing oscillations. Hence, for these model parameters, and specifically for $R_{N}=4.0$, we see the same qualitative features as without vaccination. For delay times δ well below 1, new waves come at weaker amplitudes, while for δ well above 1, the amplitudes are growing with time.

A pronounced difference, though, is the longer intervals of very low incidence between the waves in the vaccination scenarios. Assuming the time unit is approximately one week, there are about three wave peaks per year in the unvaccinated scenarios. In contrast, there is less than one wave per year if the vaccination program is implemented. The reason is that the decreasing fraction of susceptible individuals reduces the effective reproduction number, and this reduces the growth rate of the basic reproduction number during the quiet phases when the incidence is extremely low. The incidence does not start increasing again until $R_{\mathrm{eff}}\left(t\right)$ exceeds 1, and even if it then grows faster than exponential, it takes considerable time until it attains the threshold X=1 for intervention.

Another noticeable difference is that the effective reproduction number in Figure 4 remains quite low ($R_{\mathrm{eff}}<1.25$) for $t\gg \tau_{V}$. This is because $S_{V}\left(t\right)\to 0.2$ when the vaccination program is completed and $R_{\mathrm{eff}}\approx 0.2\;R$. The *basic* reproduction number R is almost five times larger and is limited by the cap $R_{N}$.

### 3.3. The Post-Pandemic Normal—A Permanent Cap on the Basic Reproduction Number

An additional message one can take home from Figure 4 is that, even when vaccination seems to have eradicated the disease, the existence of the virus in small pockets of the population is sufficient to cause strong new outbreaks if society is allowed to return to the pre-pandemic state with a large “normal” basic reproduction number $R_{N}$. In Figure 4, it is assumed $R_{N}=4.0$, which is a reasonable basic reproduction number for the most aggressive mutants circulating in early 2021 in high-income countries with no NPIs implemented.

The key to prevent this kind kind of repeated flare-up of the epidemic could be that policy makers decide to introduce some permanent NPIs that effectively introduce a long term cap $R_{N}$ on the the basic reproduction number which is lower than expected in a society with no NPIs. In practice, this would imply retaining on a permanent basis some of the most effective, but least annoying, NPIs, or invention of new NPIs. This idea, that fundamental long-term changes will have to be made in order to prevent new global outbreaks of COVID-19 or other contagious diseases, is often referred to as the new post-pandemic normal [6,7].

Figure 5 shows the results of implementing permanent NPIs giving rise to lower $R_{N}$ in the model. Specifically, it shows solutions with $R_{N}=$ 3, 2, and 1, respectively. Otherwise, the parameters are the same as in Figure 4a, where δ=1.1 and $R_{N}=4$. It is seen that in scenarios where delayed response tend to give repeated flare-ups, reduction of $R_{N}$ can slow the rise of $R_{\mathrm{eff}}\left(t\right)$ and push the second wave several years into the future. With such a long quiet period with almost no transmission of the virus one could realistically expect that a sufficient fraction of the population could be immunized from new and efficient vaccines for herd immunity to be established, or at least that the disease could enter an endemic state comparable to present day’s seasonal influenzas.

## 4. Discussion

Our model system (Equation 6) is an extremely simplified representation of a complex reality, although the second equation is probably far easier to accept than the first, since it just describes the balance between new infections produced in a population where the instantaneous reproduction number is $R_{\mathrm{eff}}\left(t\right)$ and the reduction of the number of infections due to recovery or death.The first equation, on the other hand, aspires to encapsulate the complex social dynamics that determines the evolution of R(t) in one single delay differential equation. Such a simple representation of a complex reality is certainly wrong, but may still be a valuable supplement to purely qualitative reasoning over the socio-political process that determines the response to a changing disease burden.

The potentially disastrous effect of delayed NPI response was recognized by Pei, Shandula, and Shaman (2020), employing a metapopulation transmission model and data on infections, deaths and human mobility in the United States [8]. Their findings indicate that if control measures and reductions of R(t) had been implemented just 1 to 2 weeks earlier, substantial cases and deaths could have been avoided. It is concluded that rapid detection of increasing case numbers and fast reimplementation of control measures are needed to control repeated outbreaks. They run a considerably more complex transmission model than our Equation (Equation 6), but make no attempt to model the social NPI response. They rather project the disease spread under factual and counterfactual NPI scenarios, and hence, the possibility that NPI delays may lead to a succession of epidemic waves of increasing amplitude is not explored.

The idea of NPIs trigged as incidence rates exceed a certain threshold is not new either. In a paper on strategies for mitigation and suppression of COVID-19 in countries of different income level, Walker et al. (2020) [9] modeled an oscillatory pattern of occupancy in intensive care units (ICUs) by assuming NPIs resulting in instantaneous reduction of 75% in R from a basic level 3.0 each time the threshold is exceeded, and a duration of the NPIs of 1 month. Thus, in that model R(t) flips between 0.75 and a value that starts at 3.0 but is slightly reduced in successive flips as herd immunity starts to emerge. By construction, this model will not allow the oscillations of disease incidence rate to grow in amplitude; the incidence threshold cannot be exceeded since the reproduction number drops instantaneously below 1 once the threshold is attained. The model presented here differs from [9] in two important respects:(1)The effect on R(t) of crossing the incidence threshold is not an instantaneous shift, but an effect on the rate of change $d_{t}\ln R\left(t\right)$ which is proportional to the deviation from the threshold. In this way one allows for an inertia in the response, which gives rise to a damped oscillation around the threshold incidence. This can be seen from introducing the perturbations $\tilde{R}\left(t\right)=R\left(t\right)-1$, $\tilde{I}\left(t\right)=I\left(t\right)-1$ around the equilibrium (R,I)=(1,1), and linearizing the model system (Equation 6) for δ=0, which yields the damped harmonic oscillator equation;
(10)$$d_{t}^{2}\tilde{I}+\kappa d_{t}\tilde{I}+\kappa \tilde{I}=0,$$
where the “friction coefficient” and the “spring constant” both are the same and given by κ.(2)We allow for a delay δ in the NPI-response with respect to the time the incidence threshold is crossed, and demonstrate that a sufficiently large delay may lead to a transition from a damped oscillation to a growing oscillation.

The only damping taking place in [9] is a very slow reduction in effective reproduction number arising from emerging herd immunity, i.e., reduction of the fraction of susceptible individuals, *S*. This damping effect is not taken into account in our model, since it assumed to be a negligible effect until vaccines effective against disease transmission have been rolled out in all adult age groups. On the other hand, the present work takes immunity due to vaccination into account and shows that it can prolong the quiet periods between outbreaks, but may not alone be sufficient to prevent new outbreaks altogether. Effective prevention of repeated outbreaks may require a permanent reduction of the basic reproduction number, or in other words, a societal transformation to a new post-pandemic normal. Some countries, like China, New-Zealand, and Australia, have had some success in adopting such a new normal by implementing varying approaches to reduce $R_{N}$. Common to all, though, is that this has been achieved without extensive vaccination.

The absence in [9] of delay in the social response and the absence of a reduction in the response strength due to intervention fatigue (as explored in [2]), also preclude the possibility of repeated pandemic waves of growing amplitude. Thus, the value of the scenarios depicted in [9] is primarily to quantify the fraction of the time societies will have to spend in lockdown in order to keep ICU occupancy below a given threshold, under the rather unrealistic assumption of full societal control of the reproduction number.

Under the assumption of an infectious period of $\alpha^{-1}=7$ days, and reasonable values of the parameters $\kappa_{0}$ and $R_{0}$, our model predicts that the transition from damped to growing oscillation occurs for a response delay $t_{d}$ of the order of one week. This number should of course be taken with a grain of salt, considering the simplicity of the response model. Nevertheless, it is disturbing that this critical response time turns out to be so short that it may be impossible, even under ideal circumstances, to avoid this regime of recurrent waves of growing amplitudes, provided we assume that NPIs are implemented by only using information about the most recent available case notification rate.

A valid objection to this assumption, and hence to the model system (Equation 6), is that governments have access to far more information when they lift on existing NPIs and decide on new ones. They can consider existing trends in the incidence rates, not just the the most recent reported recordings, and model projections are available. On the other hand, it is clear for anyone who follow the public debate on the necessity of interventions, that it is extremely difficult for policy makers to gain public acceptance for precautionary interventions, even in cases when a pattern has been repeated several times in the past. The common belief that the present wave is the last one seems to be a major hurdle to successful control of the epidemic.

The tendency of repeated and stronger epidemic waves has been ubiquitous across the world’s countries as we move into the second year of the pandemic. In [2], which was a precursor to the present paper, we demonstrated this by analyzing incidence data for 150 countries for the first eight months of the pandemic. Patterns of decaying as well as growing waves were observed and attempted explained by a model similar to the one presented here, but without the effects of response delay, vaccination, and a post-pandemic normal with lower $R_{N}$. In particular, repeated waves of growing amplitude were explained as a result of intervention fatigue, which was introduced in the model. Here, we show that this unstable growth may alternatively be a result of delays in the social response which only can be avoided if governments are able to take precautionary action. The key message from the present work is that relatively small changes in governments’ ability to respond in a more precautionary manner may have profound effects on controlling and mitigating new outbreaks. Modest, permanent changes towards a new post-pandemic normal can also be effective. The main challenge seems to be convey this insight to policy makers, the media, and others who shape the public opinion.

## Figures and Tables

**Figure 1 ijerph-18-04484-f001:**
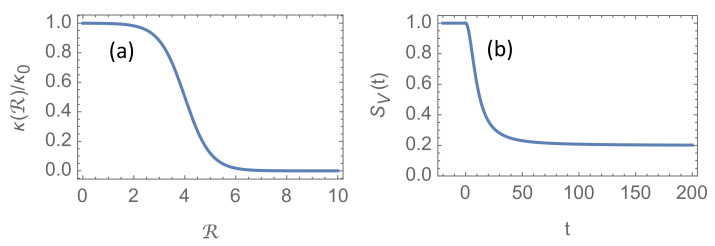
(**a**): The R-dependent coefficient $\kappa \left(R\right)/\kappa_{0}$ used in Section 3. Here the “natural” reproduction number for the dominating virus mutation has been chosen $R_{N}=4.0$. (**b**): The vaccine-induced susceptible population $S_{V}\left(t\right)$, where we have chosen $\tau_{V}=10$, corresponding to 10 weeks if we assume $\alpha^{-1}$ is 7 days.

**Figure 2 ijerph-18-04484-f002:**
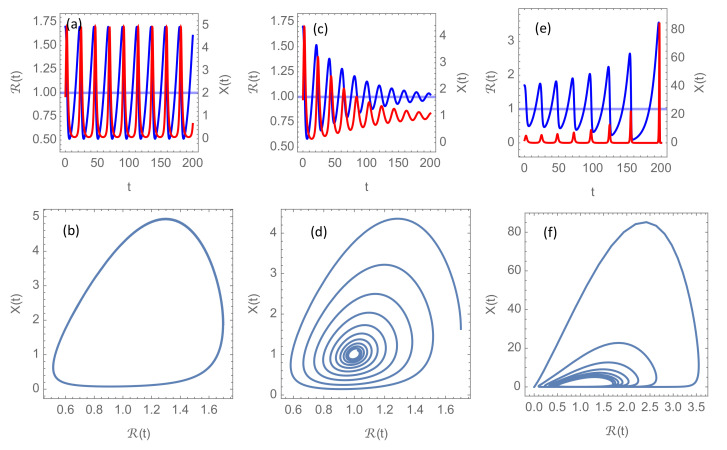
Solutions of the delay differential system (Equation 6) without vaccination ($R_{\mathrm{eff}}=R$) for initial reproduction number $R_{0}=1.7$ and normalized intervention strength $\kappa_{0}=0.1$. Blue curves show the evolution of the time-dependent reproduction number R(t) and red curves the incidence rate X(t) in units of the intervention threshold. Panels (**a**,**c**,**e**) show the graphs for R(t) and X(t) for delay times for intervention δ=0.90272,0.7, and 0.95, respectively. Times are given in units of infection duration $\alpha^{-1}$. Panels (**b**,**d**,**f**) present the corresponding phase portraits, i.e., the trajectories for the vector (R(t),X(t)) as a parameterized curve for the same values of δ.

**Figure 3 ijerph-18-04484-f003:**
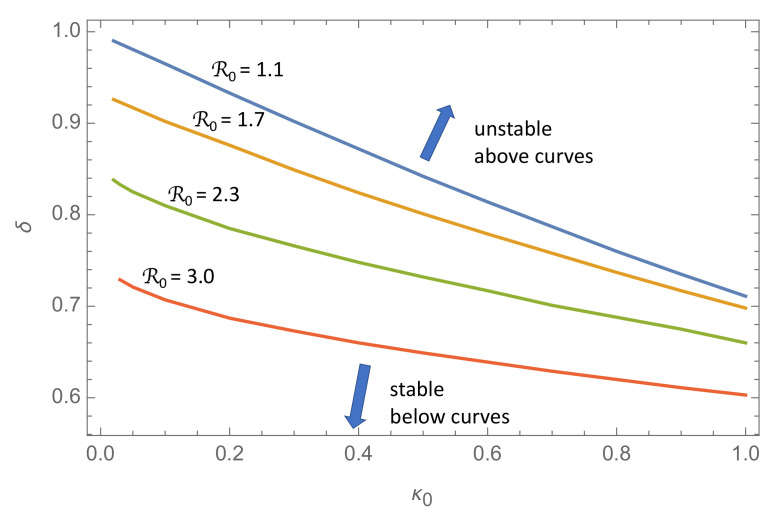
The figure depicts the transition between stable, decaying oscillations, like those shown in Figure 2c,d, and unstable, growing oscillations, as shown in Figure 2e,f. The curves represent the parameters $\kappa_{0}$ and δ at the transition points, for four values of $R_{0}$. At these transition points, the solution is a limit cycle like the one shown in Figure 2a,b. For each $R_{0}$, the unstable region of the $\left(\kappa_{0},\delta \right)$-space is the region above the corresponding curve.

**Figure 4 ijerph-18-04484-f004:**
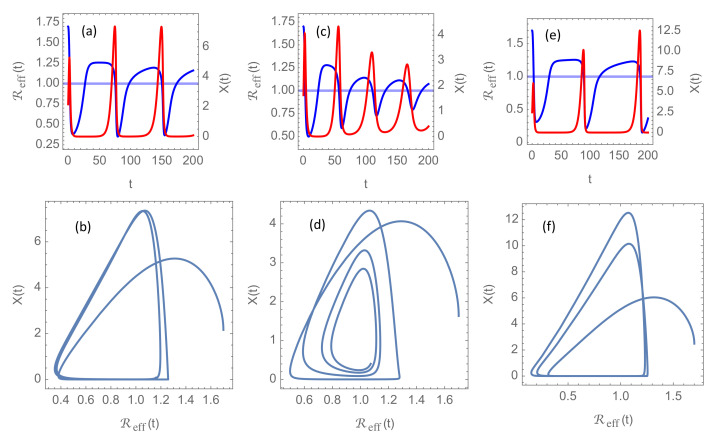
Solutions of the delay differential system Equation 6 with vaccine-induced immunity introduced through $S_{V}\left(t\right)$ given in Figure 1b, for initial reproduction number $R_{0}=1.7$ and normalized intervention strength $\kappa_{0}=0.1$. Blue curves show the evolution of the time-dependent effective reproduction number $R_{\mathrm{eff}}\left(t\right)=S_{V}\left(t\right)R\left(t\right)$ and red curves the incidence rate X(t) in units of the intervention threshold. Panels (**a**,**c**,**e**) show the graphs for $R_{\mathrm{eff}}\left(t\right)$ and X(t) for delay times for intervention δ=1.1,0.7, and 1.3, respectively. Time is given in units of infection duration $\alpha^{-1}$. Panels (**b**,**d**,**f**) present the corresponding phase portraits, i.e., the trajectories for the vector ($R_{\mathrm{eff}}\left(t\right),X\left(t\right)$) as a parameterized curve for the same values of δ.

**Figure 5 ijerph-18-04484-f005:**
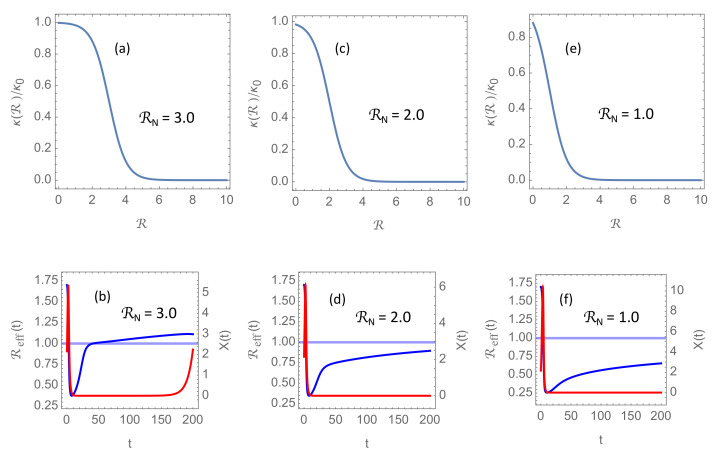
Panel (**a**,**c**,**e**) show the function $\kappa \left(R\right)/\kappa_{0}$ for $R_{N}=$ 3, 2, and 1, respectively. Panels (**b**,**d**,**f**) show solutions of the delay differential system (Equation 6) when the other model parameters are the same as in Figure 4a. Blue curves show the evolution of the effective reproduction number $R_{\mathrm{eff}}\left(t\right)$ and red curves the incidence rate X(t) in units of the intervention threshold.

## Data Availability

Data available on request.

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
