# Peer review of "The Tipping Effect of Delayed Interventions on the Evolution of COVID-19 Incidence"

_ijerph, 2021, doi:10.3390/ijerph18094484_

Round 1
Reviewer 1 Report
Major Revied

Author Response
Response to reviewer 1
Reviewer: The general feeling of your paper is towards the establishment and interpretation of purely mathematical models. But how does your mathematical model specifically prevent COVID-19 from being weak. The general feeling of your paper is towards the establishment and interpretation of purely mathematical models. But how does your mathematical model specifically prevent COVID-19 from being weak.
Response: Admittedly, this short paper formulates and explores the implications of a mathematical model. I have been I doubt about the choice of journal, and I have considered submitting it to the MPDI journal Mathematics. But mathematically, the model and the methods are standard, and there are really no new mathematical results here. The model, however, investigates the rather simple idea that delays in the implementation of NPIs will lead to higher peak incidence of an epidemic wave. Quantitative modelling is required to establish how much higher, but that is not my point in this paper. The point is to show that a sufficiently long delay may lead to a situation where new epidemic waves grow larger than their predecessors. In the revision, I have also included the effect of vaccination and the possible reduction of the basic reproduction number on a permanent basis through the establishment of a new “post-pandemic normal”.
You ask “..how does your mathematical model specifically prevent COVID-19 from being weak?” I am not sure how to understand this question, but supposing that you ask for an interpretation of the result described above, I can add the following: Most countries have implemented NPIs in direct response to rising incidence rates, and they have relaxed the NPIs in direct response to falling incidence rates. Since there is a delay from the time of infection until the decision makers receive this information, and a delay from when this information is received and action is taken, the actual incidence rate in the rising phase of a wave will grow considerably from the time an action threshold has been exceeded until action is taken. And this amplifying effect will increase with increasing delay period. This is, in fact, the main conclusion of the paper, which should be easy to grasp even if one cannot follow the mathematical detail.
Reviewer: It is highly recommended that you verify your mathematical model by looking at a country that has been successfully protected against the epidemic and using specific data.
Response: In the paper that preceded this one (Ref. [2]), we developed a model similar to the present one, without the time delay implemented, and compared results with data from the majority of the world’s countries. There it was amply documented that countries may have both damped and growing waves.
Reviewer: It is suggested that you make some graphical comparison with the literature [6] in your paper, and what advantages does your model really have compared with the literature [6].
Response: I explained in the discussion section the differences in scope and assumptions between this paper and Ref. [6]. There is nothing relevant to compare, and it has no meaning to say that one model has an advantage over the other.
Reviewer: In your Section 4, you are mainly explaining your mathematical model, but you are not doing in-depth analysis of your mathematical model in relation to epidemic prevention measures. It is suggested that your mathematical model should be combined with epidemic prevention measures, deeply analyzed and obtained some important epidemic prevention measures and important inspirations through analysis.
Response: Section 4 is the discussion section. I am sorry, but I cannot make any sense of this paragraph, and in particular not the last sentence.
Reviewer: Line 163, “...in time units of 8 days.” Many of your time do not know how to come, please give a detailed explanation and explanation, otherwise let the reader read particularly difficult. There are many such examples in the article, please explain them one by one.
Response: If I understand the reviewer correctly, he/she finds it too difficult that I use as time unit the duration of the infectious period. The reason for this is to reduce the number of free parameters in the model as much as possible. In the revision, I have explained that the reader may interpret the time unit use in the figures as one week. Without more concrete examples I cannot relate to the request to “explain them one by one”. I realize that mathematical modeling can be difficult to follow for many readers of IJERPH, but since I have already published several papers in the same style in this journal, I assume that it is within the scope of the journal.
Reviewer: Your paper has no conclusion, please add it.
Response: The most important conclusions are drawn in the Results section, but they are summarized in the last paragraph of the last section, which I have named “Discussion and conclusions.” I se no need for a separate Conclusions section.
Reviewer: In short, you have to use mathematical models to reinforce concrete practical analysis, otherwise mathematical models don't make much sense.
Response: I certainly agree that mathematical models need to relate to data, and as stated earlier, we have done extensive comparison with data in a previous paper. There is no doubt that both damped and growing oscillations in the incidence time series occur, and that incidence to a great extent is influenced by NPIs. This are the only observational facts I want to explain by this modeling. Models are useful not only for making quantitative predictions or explanation of data. In the present paper, the purpose is to point at mechanisms that can explain qualitative different developments of a pandemic. These mechanisms can be understood and discussed without mathematics, but the simple mathematical modeling reveals unexpected possible behavior that would be difficult to anticipate without this kind of modeling.
Reviewer 2 Report
This is a well-written, thoughtful analysis of the “tipping point” of delayed intervention on the evolution of COVID-19 incidence. The article will be tough going for most readers due to its mathematical content. The conclusion of the study, albeit supported by the models given their assumptions, is sobering.
They conclude that unless governments are able to respond to changes in transmission rates within approximately one week they will fail to significantly suppress a new wave of infections. In many locations this will be impossible. Given the assumptions of the models, the conclusions of the article seem valid.
My primary concern with the article is that it (admittedly) overly simplifies a very complex situation on the ground. Some of the assumptions, such as that there are no microbiological factors or changing rates of herd immunity in the model seem less justifiable as the pandemic matures.
The paper also attempt to model NPIs while ignoring the impact of the percent of the population which have received vaccines. The author does a nice job of acknowleding these limitations; however, if there were a way to test how the assumptions and relevant factors which were not considered would impact model predictions that would improve the paper’s utility for those of us involved in public health.
Also, out of curiosity, it would be illuminating to read the authors response to counter examples, such as Florida vs. California in the USA, where differences in NPI response lead to no significiant difference in incidence rates. Many in the public health literature have concluded that the situation is so complex (e.g., housing density, number of densely populated cities, differences in income levels, age, and race, etc.) that modeling the impact of NPI alone is insufficient to provide actionable data.
Author Response
Response to reviewer 2
The substantial revision and expansion of the paper revision has taken into account the concerns expressed by the reviewer in the following ways:
The possibility of changing basic reproduction number due to microbiological effects such as new mutants and post-pandemic, permanent NPIs has been taken into account through the modeling of a cap RN on this number.
The effect of immunity due to vaccines has now been incorporated in the model, and some interesting conclusions have been drawn from the simulations.
I haven’t complied with the reviewer’ desire to show counter examples to the effect of NPIs. I am sure that these are numerous, but at least the experience from Europe, which is what I am most familiar with, is that NPIs work, but in some cases, they are counteracted by factors like the emergence of new mutants, intervention fatigue in the population, or political sabotage. It would be far beyond the scope of this study to embark on a discussion of such complications. Comparison of a model similar to this one, without the time delay, with incidence data from most countries in the world was made in our previous paper, Ref [2].
Round 2
Reviewer 1 Report
Your paper needs revision

Author Response
As mentioned in my first response, we made extensive comparisons with real-world data in reference [2], which was a precursor to the present paper. I have added the two following paragraphs in the Discussion section:
Some countries, like China, New-Zealand, and Australia, have had some success in adopting such a new normal by implementing varying approaches to reduce RN. Common to all, though, is that this has been achieved without extensive vaccination.
The tendency of repeated and stronger epidemic waves has been ubiquitous across the world's countries as we move into the second year of the pandemic. In [2], which was a precursor to the present paper, we demonstrated this by analyzing incidence data for 150 countries for the first eight months of the pandemic. Patterns of decaying as well as growing waves were observed and attempted explained by a model similar to the one presented here, but without the effects of response delay, vaccination, and a post-pandemic normal with lower RN. In particular, repeated waves of growing amplitude were explained as a result of intervention fatigue, which was introduced in the model. Here, we show that this unstable growth may alternatively be a result of delays in the social response which only can be avoided if governments are able to take precautionary action.